# Influence of a Hydrocarbon Biodestructor on the Growth and Content of Phytohormones in *Secale cereale* L. Plants under Petroleum Pollution of the Soil

Yulia Sotnikova [1], Anna Grigoriadi [1], Vadim Fedyaev [1], Margarita Garipova [1], Ilshat Galin [2], Guzal Sharipova [2], Anna Yamaleeva [1], Sergey Chetverikov [3], Dmitriy Veselov [2], Guzel Kudoyarova [2,*] and Rashit Farkhutdinov [1]

1   Biological Department, Ufa University of Science and Technology, 450076 Ufa, Russia; sotnikova-bashedu@mail.ru (Y.S.); nysha111@yandex.ru (A.G.); vadim.fedyaev@gmail.com (V.F.); margaritag@list.ru (M.G.); yamaleev3a@yandex.ru (A.Y.); frg2@mail.ru (R.F.)
2   Laboratory of Plant Physiology, Ufa Institute of Biology, Ufa Federal Research Centre, Russian Academy of Science, 450054 Ufa, Russia; ilshat.rafkatovitch@gmail.com (I.G.); g.v.sharipova@mail.ru (G.S.); veselov@anrb.ru (D.V.)
3   Laboratory of Agrobiology, Ufa Institute of Biology, Ufa Federal Research Centre, Russian Academy of Science, 450054 Ufa, Russia; che-kov@mail.ru
*   Correspondence: guzel@anrb.ru

**Abstract:** The phytoremediation of soil contaminated with petroleum depends on the association of plants with rhizosphere bacteria capable of promoting plant growth and destroying petroleum hydrocarbonates. In the present work, we studied the effects of "Lenoil" biological product containing bacteria *Pseudomonas turukhanskensis* IB 1.1, capable of destroying petroleum hydrocarbons on *Secale cereale* L plants, which previously proved to be resistant to the weak oil pollution of gray forest soil and to the composition of microorganisms in their rhizosphere. The composition of microorganisms in the rhizosphere of rye roots was studied, morphometric parameters of shoots and roots of rye plants were estimated, and hormone concentration was immunoassayed under conditions of 4% petroleum pollution of the soil. Addition of petroleum to the soil increased the content of oligonitrophilic (by 24%) and hydrocarbon-oxidizing (by 33%) microorganisms; however, the content of cellulolytic (by 12.5 times) microorganisms in the rhizosphere decreased. The use of Lenoil led to a further increase in the number of cellulolytic (by 5.6 times) and hydrocarbon-oxidizing (by 3.8 times) microorganisms and a decrease in the number of oligonitrophilic (by 22.7%) microorganisms in the rhizosphere. Under petroleum pollution, the content of auxins (IAA), zeatin riboside, zeatin nucleotide, and zeatin decreased, while the content of abscisic acid (ABA) increased in the shoots of rye plants. Lenoil treatment led to an eight-fold increase in the IAA content in the roots and a decrease in the ABA content in the aerial part and in the roots. It was shown for the first time that the treatment of petroleum-contaminated soil with "Lenoil" increased root mass due to the development of lateral roots, concurrent with high root IAA content. Petroleum pollution increased the deposition of lignin and suberin in the roots, which strengthened the apoplastic barrier and, thus, reduced the infiltration of toxic components. The deposition of suberin and lignin decreased under "Lenoil" treatment, indicating a decrease in the concentration of toxic petroleum components in the soil degraded by the bacteria. Thus, the biological preparation reduced the growth-inhibiting effect of petroleum on rye plants by increasing the content of growth-stimulating phytohormones (IAA and cytokinins) and reducing the content of ABA, justifying the importance of further study of relevant hormones for the improvement of phytoremediation.

**Keywords:** *Secale cereale* L.; petroleum contamination of the soil; "Lenoil" biopreparation; phytoremediation; phytohormones; lignin; suberin

## 1. Introduction

Petroleum pollution changes the structure and geochemical properties of the soil, reduces seed productivity and seed germination rate, inhibits growth and disturbs plant development, and changes the number and qualitative composition of soil microorganisms [1,2]. In soil contaminated with petroleum, a selection of microorganisms most resistant to this type of pollution takes place. The higher susceptibility of the plant root system to petroleum compared with the aboveground part is explained both by the direct effect of pollutants on the roots and the consequence of changes in the soil structure and rhizosphere microorganisms interacting with plants [1].

To organize the purification of soils contaminated with petroleum products by phytoremediation, it is necessary to properly select plants that are maximally adapted to certain soil and climatic conditions and the level of pollution [3,4]. It is known that the use of microbial preparations based on different types of microorganisms can increase plant resistance and stimulate plant growth under adverse conditions due to the synthesis of phytohormones (auxins and cytokinins) performed by them [5–7].

To achieve efficient phytoremediation, plants should have a high rate of biomass production, the ability to absorb and accumulate heavy metals and other xenobiotics in their tissues, and also increase the activity of soil microorganisms [8].

Therefore, when evaluating the effectiveness of microbiological preparations intended for the biodegradation of petroleum products, it is necessary to study the complex of anatomical and physiological parameters of phytoremediators [9,10]. We are interested in plant–microbial interaction under conditions of plant growth under soil contamination.

The aim of this research is to study the effects of the biological preparation "Lenoil" containing bacteria of the *Pseudomonas turukhanskensis* IB 1.1 strain on the quantitative and qualitative composition of various groups of microorganisms in the rhizosphere as well as the growth balance of phytohormones under conditions of soil petroleum pollution.

## 2. Materials and Methods

### 2.1. Plant Growth Conditions

The experiments were carried out under laboratory conditions using grey forest soil collected in the northeastern part of the Ufa district of the Republic of Bashkortostan. The soil was ground up and sifted through 3 mm mesh. The soil had the following agrochemical indicators: clay-illuvial agro-chernozem, characterized by medium humus content (4–5%) and slight acid reaction (pH $5.6 \pm 0.4$).

The residual petroleum content in the soil was determined as described [11].

The composition of the biopreparation "Lenoil"®—NORD, SHP (CJSC NPP "Biomedkhim", Ufa, Russian Federation) includes the bacteria *Pseudomonas turukhanskensis* IB 1.1 (titer not less than $1 \times 10^8$ CFU g$^{-1}$) [2].

Five plants of *Secale cereale* L. (cv Tatiana) were grown in 0.5 L pots with a 12 h light period, illumination of 535 µmol s$^{-1}$ m$^{-2}$ PAR, and an air temperature of 22–25 °C. Soil moisture was maintained throughout the experiment at 60% of full capacity. Commercial petroleum was added to the soil at a concentration of 4% of the dry weight of the soil. After 30 days, 0.3 mL of "Lenoil" per 100 g of dry soil was added to half of the vessels containing petroleum. Plants were sown 30 days after the addition of the biopreparation. Plants grown in soil not contaminated with petroleum served as control.

### 2.2. Soil Microorganism Evaluation

The total number of microorganisms in the rhizosphere of rye roots in the average soil sample was taken into account, which was compiled by mixing 10–15 separate samples dug together with plants.

The soil adhering to plant roots was shaken off onto a sterile glass, and the adhering particles were removed with a sterile scalpel [12].

To estimate the abundance of microorganisms—heterotrophs, cellulolytics, oligonitrophils, micromycetes, and hydrocarbon-oxidizing microorganisms (HOM)—they were

cultivated in vitro on the respective media: meat-peptone agar (MPA), Getchinson, Ashby, Chapek, and Dianova–Voroshilova [13].

### 2.3. Hormone Extraction, Purification, and Immunoassay

Extraction, purification, and determination of auxins (IAA) and abscisic acid (ABA) content in plant material by enzyme-linked immunosorbent assay were carried out as described [14,15]. Purification and analysis of the content of cytokinins (zeatin, its riboside, and nucleotide) were performed according to [16,17].

### 2.4. Detection of Lignin and Suberin Localization

To study changes in the anatomy of rye root tissues, freehand transverse sections were prepared from the basal part of the plant roots.

Lignin and suberin were detected as described in [18]: the sections were stained with an aqueous solution of berberine hemisulfate (0.1% $w/v$) for 1 h and washed twice with distilled water; then, sections were counterstained for 15 min with toluidine blue (0.05% $w/v$) in 0.1 M phosphate buffer (pH 5.6) to enhance the fluorescence intensity, washed twice with distilled water, embedded in a mixture of 0.1% $FeCl_3$/50% glycerol, and covered with a coverslip. To study the sections, an Olympus FluoView FV3000 laser scanning confocal microscope (Olympus, Tokyo, Japan) was used.

### 2.5. Root Morphology Assessment

To assess the morphological changes in the root system structure, the roots were washed from the soil with water, placed in a 50% Hoagland–Arnon nutrient solution, and analyzed at $100\times$ magnification using an Axio Imager.A1 light microscope equipped with an AxioCam MRc5 digital camera (Carl Zeiss AG, Oberkochen, Germany).

### 2.6. Statistics

The data were statistically processed using standard MS Excel program. The tables and figures show means and their standard errors (SEs). The significance of differences was assessed by ANOVA followed by Duncan's test ($p \leq 0.05$).

### 3. Results and Discussion

#### 3.1. Composition of Microorganisms in the Rhizosphere of Rye Roots

Application of the "Lenoil" microbiological preparation decreased petroleum content in the soil from 4 to 2.4% after the first 30 days without plants; after another 30 days in the presence of plants, it decreased to 1.6%. In the variant without soil treatment with the preparation, petroleum content was found to be 2.8% after 2 months of the experiment.

Our assessment of the indicator of the state of soil microbiocenosis showed that in the control variant, the total number of heterotrophic microorganisms was about $9 \cdot 10^6$ CFU $g^{-1}$ of soil in the rhizosphere of rye plants. Under the influence of the pollutant, this indicator decreased by 63%, which could be due to the direct toxic effect of petroleum hydrocarbons.

The use of the "Lenoil" preparation increased the number of heterotrophic microorganisms up to about $3 \cdot 10^7$ CFU $g^{-1}$ of soil (compared with $3 \cdot 10^6$ CFU $g^{-1}$ of contaminated soil). This group of microorganisms characterizes the total microbial number, and the change in the number depends on the proportion of sensitive representatives of the microbiome both in whole soil and in the plant rhizosphere [19,20]. It is known that the treatment with biopreparation improves the living conditions of oil-contaminated soils for both plants and rhizospheric microorganisms [2]. The results of the present research are in accordance with literature data indicating that the use of microbial preparations for the remediation of petroleum-contaminated soils affects the quantitative composition of the native microbiota, including in the rhizosphere of phytoremediation plants [2,21].

There was also a significant growth in microscopic fungi, both in petroleum-contaminated and "Lenoil"-treated soil. In the presence of petroleum in the rye rhizosphere, the number of micromycetes increased by about 20 times compared with the control (from $1 \cdot 10^3$ to

$22 \cdot 10^3$ propagules g$^{-1}$ of soil), and after adding the biological product, the increase was two times greater (up to $46.75 \cdot 10^3$ propagule g$^{-1}$ soil). This can be explained by the resistance of many microscopic fungi to the influence of petroleum and their ability to de-grade it [22,23]. Microorganisms that are constantly present in the rhizosphere microbiota include representatives of oligonitrophils (ONPh) and cellulolytics (CLB) [24]. Figure 1 shows that the number of cellulolytic microorganisms was significantly reduced under the influence of petroleum because this group of microorganisms is highly sensitive to the action of this pollutant [2,22]. The addition of hydrocarbon-oxidizing bacteria (HOM) of the "Lenoil" preparation had a positive effect on the number of cellulolytic bacteria; how-ever, the toxic effect of petroleum was not completely eliminated and their number was not restored to control values (Figure 1).

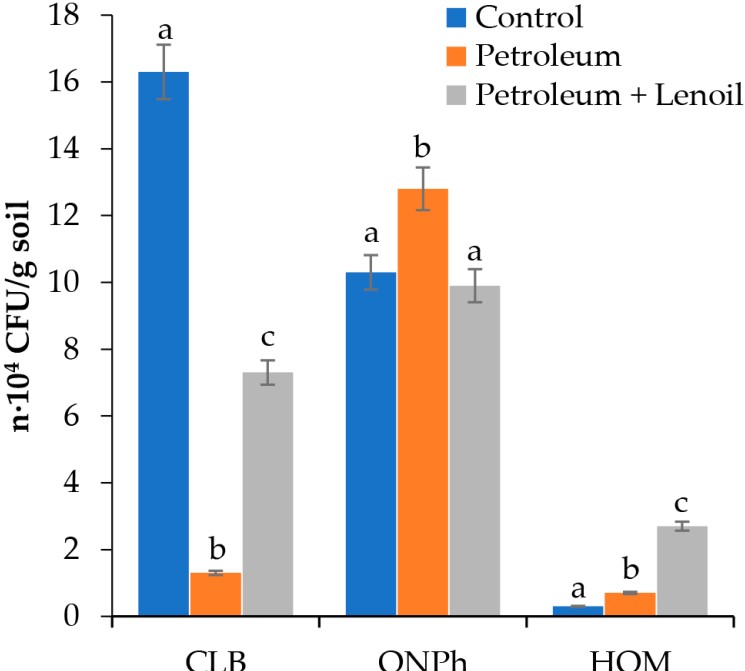

**Figure 1.** The number of cellulolytic bacteria (CLB), oligonitrophilic bacteria (ONPh), and hydrocarbon-oxidizing microorganisms (HOM) in the rhizosphere of *Secale cereale* L. plants in the control variant (Control) under conditions of petroleum pollution of the soil (Petroleum) and under using the "Lenoil" preparation (Petroleum + Lenoil). Data are shown as mean $\pm$ SE. Significantly different means are marked with different letters at $p \leq 0.05$ (ANOVA followed by Duncan's test).

There were no significant changes in the number of oligonitrophils (nitrogen-fixing anaerobic microorganisms) in the presence of oil contamination of the soil and biological product (Figure 1), which indicates the stability of these microorganisms.

Hydrocarbon-oxidizing microorganisms belong to a specific group of microorganisms. Their natural presence in the soil has been described; however, the addition of petroleum hydrocarbons, which are their substrates, leads to an increase in their abundance [2,25], which is consistent with our data (Figure 1).

After application of "Lenoil", maximum values of the number of HOM were detected in the rhizosphere of rye plants. In this variant of the experiment, the growth of this group of microorganisms was stimulated both due to plant–microbial interaction [21,26,27] and the introduction of HOM in the form of a biological product.

Thus, the use of the "Lenoil" preparation for the treatment of petroleum-contaminated soil and phytoremediation plants stimulated the increase in the number of hydrocarbons degrading microorganisms and contributed to the maintenance of the number of cellulolytic and oligonitrophilic bacteria in the rye rhizosphere.

*3.2. Morphometric Parameters of Shoots and Roots of Rye Plants under Soil Pollution and "Lenoil" Treatment*

It seemed interesting to study the influence of the "Lenoil" preparation on the growth response and the morphological structure of the roots of rye plants under soil petroleum pollution. Under such conditions, the morphometric parameters of shoots and roots decreased (Table 1). The use of the "Lenoil" biopreparation led to an increase in the length of shoots by 1.8 times compared with the control plants and by 3.8 times in comparison with plants untreated with "Lenoil" and growing on petroleum-contaminated soil. Under petroleum contamination, the biopreparation increased the accumulation of fresh mass of shoots by 4.4 times and was greater than that in plants of the control variant by 1.3 times (Table 1).

**Table 1.** Comparative evaluation of the morphometric parameters of *Secale cereale* L. plants under conditions of soil pollution with petroleum and with the use of the "Lenoil" preparation.

| Plant Part | Parameter | Control | Petroleum | Petroleum + "Lenoil" |
|---|---|---|---|---|
| Shoot | Length, cm | 17.83 [a] ± 1.22 | 8.46 [b] ± 0.52 | 32.11 [c] ± 1.32 |
| | Fresh weight, g | 8.36 [a] ± 0.66 | 2.38 [b] ± 0.26 | 10.46 [a] ± 0.96 |
| Root | Length, cm | 12.24 [a] ± 1.04 | 7.34 [b] ± 0.56 | 18.41 [c] ± 1.26 |
| | Fresh weight, g | 1.75 [a] ± 0.12 | 0.47 [b] ± 0.04 | 3.82 [c] ± 0.22 |

Data are shown as mean ± SE. Significantly different means are marked with different letters at $p \leq 0.05$ (ANOVA followed by Duncan's test).

Comparison of the length and fresh weight of the "Lenoil"-treated roots with the control variant and plants under petroleum pollution untreated with "Lenoil" also showed a stimulating effect of the treatment (Table 1). It should be noted that the morphometric parameters of plants treated with the preparation were higher than in the plants of the control variant, although the former were grown in contaminated soil.

We also evaluated the following indicators of the root system: the total length of the roots and the number of primordial and lateral roots (Table 2). The study showed that the total length of the roots was greater in the control plants and in those treated with "Lenoil" compared with the untreated plants grown in contaminated soil, which confirms the stimulation of the root system development of rye plants under the influence of the biological preparation (Table 2).

**Table 2.** Changes in the morphological structure of the roots of 30-day-old *Secale cereale* L. plants under conditions of soil contamination with petroleum and treatment with "Lenoil" preparation.

| Experiment Variant | Total Length of Roots, cm | Number of Primordia, Pixels/Root | Number of Lateral Roots, Pixels/Root |
|---|---|---|---|
| Control | 17.22 [b] ± 0.28 | 7.1 [a] ± 0.5 | 3.2 [c] ± 0.2 |
| Petroleum | 8.26 [a] ± 0.68 | 11.7 [b] ± 0.3 | 0.5 [a] ± 0.2 |
| Petroleum + Lenoil | 24.78 [c] ± 0.36 | 10.8 [b] ± 0.4 | 2.1 [b] ± 0.1 |

Data are shown as mean ± SE. Significantly different means are marked with different letters at $p \leq 0.05$ (ANOVA followed by Duncan's test).

*3.3. Concentration of Hormones in Rye Plants under Soil Pollution and Treatment with "Lenoil"*

The relatively low growth rate of the aerial parts and roots of plants under soil petroleum contamination (Table 1) corresponded to the low content of auxins (IAA) in the plant shoots and roots (Figure 2). In control plants, the maximum content of IAA was observed in the shoots. Under soil pollution, the use of "Lenoil" increased IAA concentration both in roots and shoots, and the highest level of IAA was found in these roots, which could stimulate the growth of roots in length and the accumulation of their biomass [28].

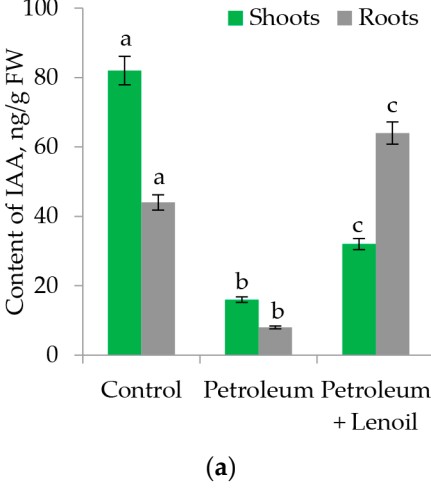

(**a**)

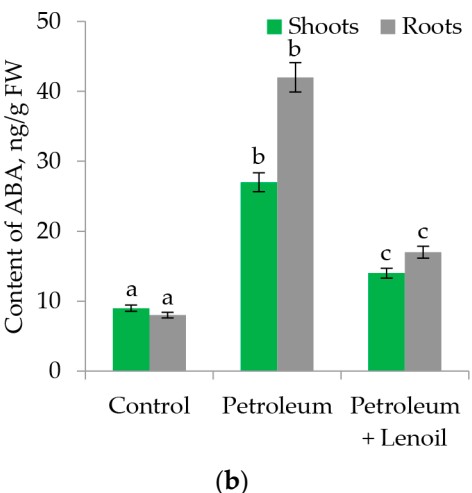

(**b**)

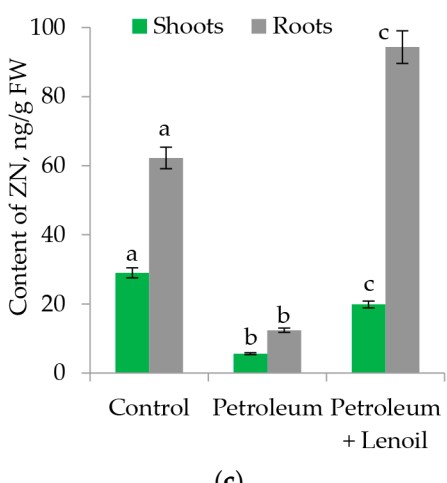

(**c**)

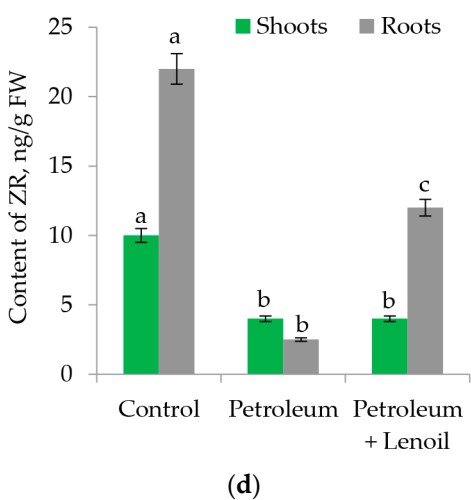

(**d**)

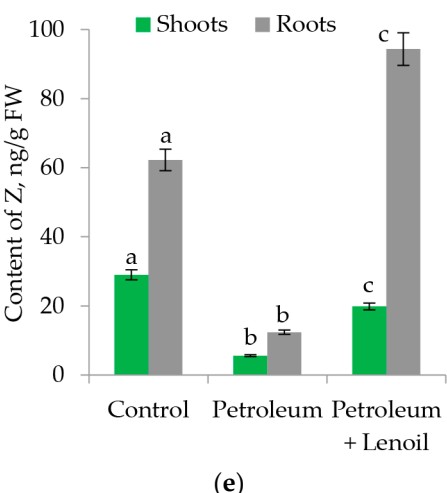

(**e**)

**Figure 2.** The content of IAA (**a**), ABA (**b**), zeatin nucleotide (**c**), zeatin riboside (**d**), and zeatin (**e**) in the shoots and roots of rye plants in control (Control) under conditions of petroleum pollution of the soil (Petroleum) and under the "Lenoil" preparation (Petroleum + Lenoil). Data are shown as mean ± SE. Significantly different means are marked with different letters at $p \leq 0.05$ (ANOVA followed by Duncan's test).

Previously, the "Lenoil" producers found that the hydrocarbon-oxidizing microorganisms of the *Pseudomonas* genus are capable of synthesizing significant concentrations of IAA and cytokinins [2]. Our results show that the ability of microorganisms to synthesize plant hormones can influence the content of these hormones in planta. It is known that phytohormones synthesized by microorganisms of biological preparations can affect the processes of plant adaptation to environmental conditions and form one of the factors enabling plant resistance [29]. It is important that, along with other mechanisms, artificial symbiotic associations of plants and microorganisms are established in oil-contaminated soil due to the mutual exchange of phytohormones between them [7,30].

Determination of abscisic acid (ABA) content in rye plants growing on soil with petroleum showed that concentration of this hormone in shoots and roots was higher compared with the control (Figure 2). This could lead to the inhibition of shoot growth under these conditions of soil pollution [6,31]. It should be noted that the use of the biopreparation "Lenoil" led to a decrease in ABA content in the shoot down to the values close to those obtained for plants grown on uncontaminated soil, which indicates a decrease in the negative effect of soil pollution under the influence of the biopreparation (Figure 2b).

It is known that the accumulation of ABA in roots in response to stress usually leads to a decrease in the growth processes of plant roots [32]. In some cases, the accumulation of ABA in the roots can lead to an increase in the root hydraulic conductivity to compensate for water absorption problems that occur in petroleum-contaminated soil; thus, plants can support growth processes in the shoot and root [33].

We found changes in the content of zeatin derivatives, which, as is known from the literature, play a different role in adaptation to adverse conditions [34–36]. Determination of the content of zeatin nucleotide (ZN) in rye plants showed that its content in the shoots and roots of control plants was higher than in plants under petroleum pollution (Figure 2c). The presence of the "Lenoil" preparation in petroleum-contaminated soil led to an increase in ZN content in the aerial part and its accumulation in the roots of rye plants, which indicates inhibition of the loading of this reserve form of cytokinins into the shoots (Figure 2c) [34,35].

The content of zeatin riboside (ZR) in the shoots of rye plants of the control variant was higher compared with the values of the ZR content in plants grown under conditions of petroleum pollution; in the root system, the content of ZR exceeded the content of the hormone in the shoot by almost two times (Figure 2d). When the soil was treated with the "Lenoil" preparation, as in the control variant, inhibition of the transport of ZR to the aerial part was observed, which was manifested in the accumulation of the hormone in the roots (Figure 2d). As is known from the literature, ZR transport depends on the rate of formation of conjugated derivatives at the site of synthesis and the physiological state of the plant [35]. Consequently, under the action of the biological preparation, activation of the synthesis of ZR occurred; however, the content of the transport form of cytokinins in the shoot was approximately the same as in polluted plants untreated with Lenoil (Figure 2d), which could be associated with a decrease in the level of loading of the transport form of cytokinins into the phloem stream [36].

Comparison of zeatin (Z) content in rye plants of the control variant and plants grown under petroleum pollution of soil showed that zeatin was redistributed to the aerial part in the control variant; further, in the case of the pollution without "Lenoil" treatment, this form of cytokinins accumulated in the roots (Figure 2e). When the petroleum-contaminated soil was treated with the "Lenoil" preparation, the content of Z in the roots and especially in the shoots increased (Figure 2d), and the absolute values exceeded the parameters of the plants of the control variant by more than two times. According to the literature, an increase in Z content can lead to the activation of growth by cell division in the shoot and an increase in resistance to a stress factor [36], which, in our work, was manifested as an increase in the morphometric parameters of plants treated with "Lenoil" (Table 1).

*3.4. Root Structure of Rye Plants under Soil Pollution and Treatment with "Lenoil"*

The study of the root structure using light microscopy showed that under petroleum pollution, the highest number of primordia was formed in plants (Table 2) but most of them did not develop into lateral roots by the age of 30 days. This could be due to the low content of IAA (Figure 2a) in the roots of petroleum-contaminated plants and the accumulation of ABA in them (Figure 2b). Auxin is known to be involved in the development of root primordia and their subsequent development [37], while a high level of ABA inhibits their growth [38,39].

The treatment of petroleum-contaminated soil with "Lenoil" led to an increase in the number of lateral roots in rye plants accompanied by a high level of IAA in the roots (Figure 2a) and redistribution of zeatin to the aerial part at the same time (Figure 2e). It is known that the microorganisms that make up the "Lenoil" biopreparation are auxin producers and have a growth-stimulating effect on plant root formation under conditions of soil petroleum pollution [30]. It can be assumed that the increased branching rate in "Lenoil"-treated plants is associated with the simultaneous synthesis of phytohormones by plants and microorganisms located in the rhizosphere.

The absorption zone with well-developed root hairs is clearly visible in the photographs of the roots of plants of the control variant (Figure 3a). The presence of petroleum in the soil obviously led to a certain physical effect on the root hairs when they lost their "fluffy" structure (Figure 3b). Figure 3b shows that under "Lenoil" treatment, the appearance of the differentiation zone with root hairs was similar to that of the control plants. It is known that some microorganisms of biological products can eliminate the petroleum film from the surface of root hairs with the help of biosurfactants [2,40,41].

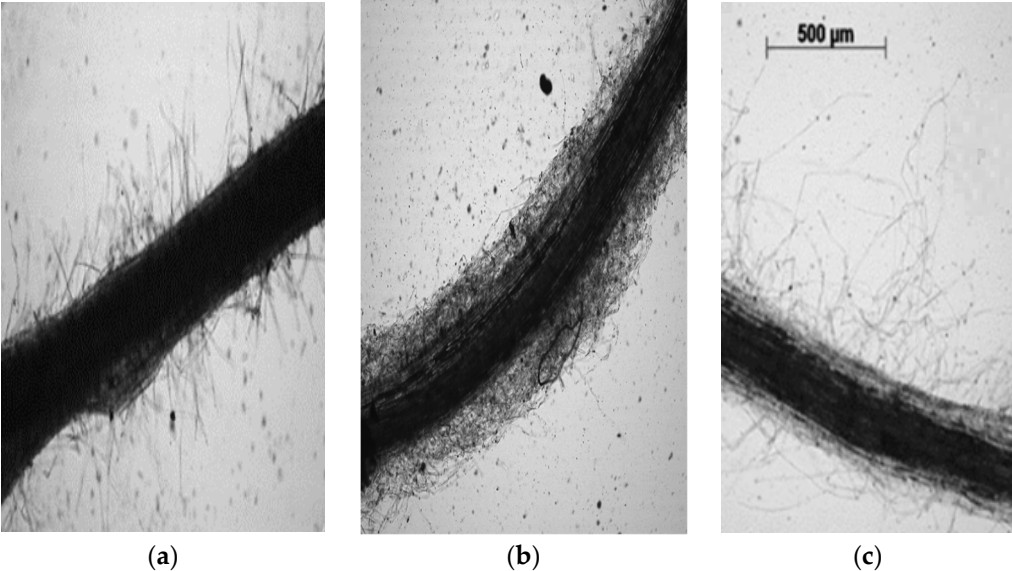

(**a**)　　　　　　　　　　　　　　(**b**)　　　　　　　　　　　　　　(**c**)

**Figure 3.** Photograph (×100) of the root hair zone of 30-day-old rye plants grown (**a**) in soil without petroleum, (**b**)—in soil containing 4% petroleum, and (**c**)—in soil containing 4% petroleum, after applying the biopreparation "Lenoil".

Thus, rye plants treated with "Lenoil" preparation had a more developed root system and a higher number of developing lateral roots compared with untreated plants grown in contaminated soil and with a normalized structure of the root hair zone (Tables 1 and 2). This may also indicate the presence of a symbiotic microbial–plant association that successfully adapts to adverse environmental conditions.

Determining lignin and suberin localization on transverse sections of the basal part of the roots revealed an increase in their deposition under the influence of petroleum (Figure 4f), which may indicate the formation of apoplastic barriers in plant roots to protect them against the penetration of toxic substances contained in the petroleum.

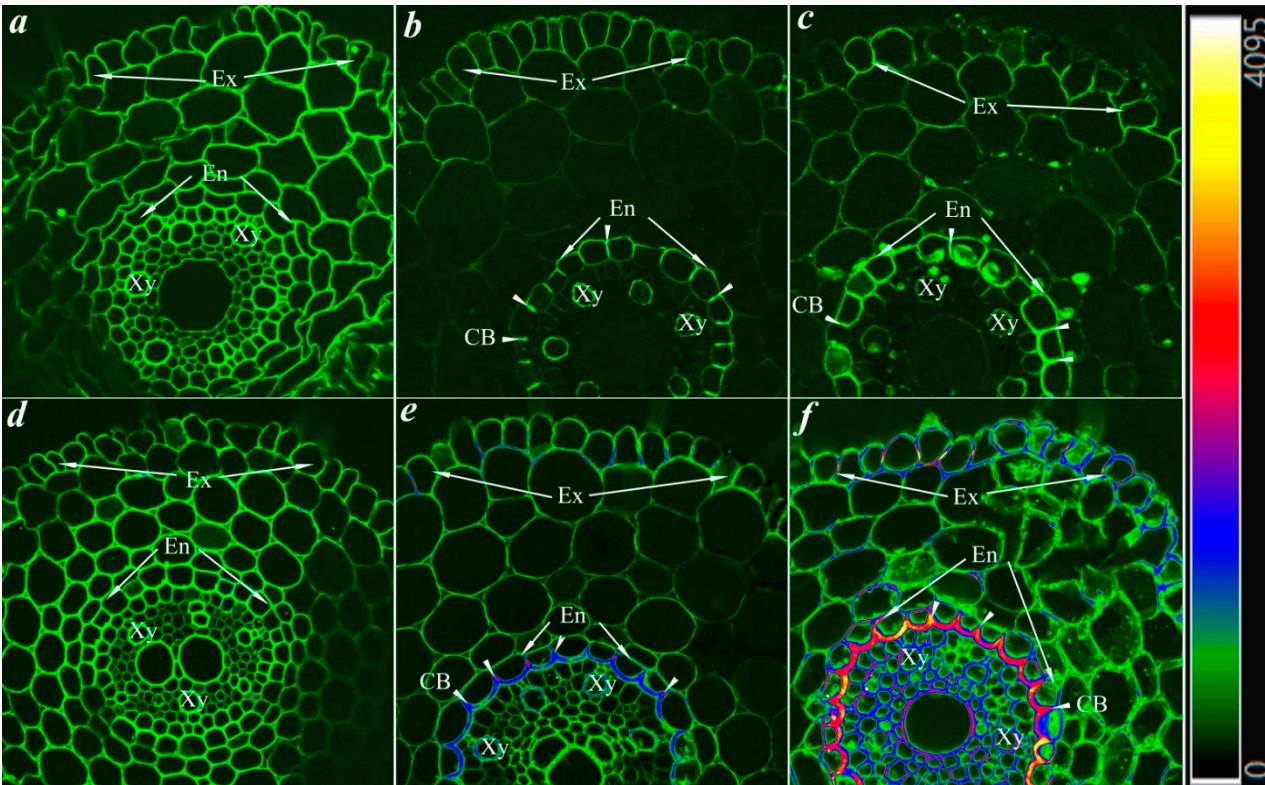

**Figure 4.** Photos (×100) of the localization of lignin and suberin and detection of Casparian bands in root cross-sections of the 30-day-old rye plants untreated (**a**,**d**), treated with "Lenoil" (**b**,**e**), and treated with petroleum (**c**,**f**) from the segments of the basal parts (**a**–**c**) of the roots and parts, located 2 cm (**d**–**f**) from the root apex. CB—Casparian bands (shown by arrowheads), En—endoderm, Ex—exoderm, Xy—xylem. The heatmap shows color-coded fluorescence signal intensities of laser.

The use of the "Lenoil" microbial preparation in the case of soil petroleum contamination reduced the content of toxic substances, which apparently led to a decrease in the accumulation of lignin and suberin (Figure 4e). It is known that the deposition of lignin in the region of the radial walls of the endoderm of the Caspari strips leads to an increase in apoplast barriers; however, this also reduces the hydraulic conductivity of the roots, which can inhibit the growth of the plant shoots [42].

Thus, the use of the "Lenoil" biopreparation in conditions of soil petroleum pollution reduced the content of substances causing lignification of rye roots and stimulated the growth of the root system (Tables 1 and 2). Growth processes were apparently controlled by the hormonal system; so, inhibition of the growth of rye plants on oil-contaminated soil likely occurred due to the high content of ABA in the tissues of the shoot and root system and the relatively low concentrations of IAA and various forms of zeatin. The addition of the petroleum biodestructor "Lenoil" to the soil increased the content of IAA in the roots and zeatin in the shoots, thereby activating growth processes.

## 4. Conclusions

The results of the present research are important for understanding plant–microbial interactions and using the results in the restoration of petroleum-contaminated soils with phytoremediants and bacterial preparations. The application of the microbiological preparation "Lenoil" containing hydrocarbon-oxidizing microorganisms of the *Pseudomonas* genus led to an amelioration of living conditions for rye plants resulting in improved morphometric parameters of rye plants. The effects of "Lenoil" can be not only due to the presence of *Pseudomonas* microorganisms in the product itself but also due to its effect on the composition of microorganisms in the soil. The concentration of auxins and cytokinins

increased in plants treated with Lenoil in accordance with the previously shown ability of microorganisms of this biological product to synthesize these hormones [2]. There was a decrease in the content of the stress phytohormone, ABA, both in the aerial part and in the roots of "Lenoil"-treated plants. The content of IAA increased in both parts of the plant but to a greater extent in the roots, which stimulated their growth. The rapid increase in the length of the aerial part was associated with an increase in zeatin content. In plants of this variant, the lengths of the aerial parts and roots were close to the values of the control variant grown in the absence of petroleum contamination. Thus, under the conditions of petroleum pollution and soil treatment with "Lenoil", we showed the effect of the biological product on the state of the endogenous hormonal system of plants, which led to the activation of growth processes and a change in the structure of the root cell wall. Thus, estimation of the content of IAA, cytokinins, and ABA is important for selecting effective concentrations of biological preparations and creating their combinations during phytoremediation activities.

**Author Contributions:** Conceptualization, D.V., G.K. and R.F.; methodology, Y.S., A.G. and R.F.; validation, A.Y. and G.K.; formal analysis, Y.S., A.G., M.G. and R.F.; investigation, Y.S., A.G., V.F., I.G. and G.S.; data curation, G.K., M.G. and R.F.; writing—original draft preparation, Y.S., G.K. and R.F.; writing—review and editing, A.G., V.F., G.K. and R.F.; visualization, A.G., V.F., I.G., R.F. and G.S.; supervision, G.K. and R.F.; project administration, S.C., D.V., G.K. and R.F.; funding acquisition, R.F. All authors have read and agreed to the published version of the manuscript.

**Funding:** The study was funded by the Russian Science Foundation, grant number 23-24-00358.

**Institutional Review Board Statement:** Not applicable.

**Informed Consent Statement:** Not applicable.

**Data Availability Statement:** Not applicable.

**Conflicts of Interest:** The authors declare no conflict of interest. This study used biopreparation "Lenoil"®—NORD, SHP (CJSC NPP "Biomedkhim", Ufa, Russian Federation) and hereby declares that there are no conflicts of interest.

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
