# Peer review of "Influence of a Hydrocarbon Biodestructor on the Growth and Content of Phytohormones in Secale cereale L. Plants under Petroleum Pollution of the Soil"

_agriculture, doi:10.3390/agriculture13081640_

Round 1
Reviewer 1 Report
1. The total number of words in the article is (word count): 6558 words, 2 tables, 4 figures.
Table 2. Comparative evaluation of the morphometric parameters of Secale cereale L. plants under 180 conditions of soil pollution with petroleum and with the use of the "Lenoil" preparation à italicize the name of the species
Table 2. Changes in the morphological structure of the roots of 30-day-old rye plants under condi- 198 tions of soil contamination with petroleum and treatment with "Lenoil" preparation è need to renumber these 2 tables
2. Title: “Influence of a Hydrocarbons Biodestructor on the Growth and 2 Content of Phytohormones in Secale Cereale L. Plants under Oil 3 Pollution of the Soil”
--> The capitalization of the first character of a letter is a rule of the journal or not, if it is a rule of the journal, it is okay, but if it is not the rule of the journal, the author should limit the capitalization of first characters in the title of the article.
3. "1 Biological Department, Ufa University of Science and Technology, 450076 Ufa, Russia; sotnikova- 7 bashedu@mail.ru (Y.S.); nysha111@yandex.ru (A.G.); vadim.fedyaev@gmail.com (V.F.); margaritag@list.ru 8 (M.G.); yamaleev3a@yandex.ru (A.Y.); frg2@mail.ru (R.F.) 9
2 Laboratory of Plant Physiology, Ufa Institute of Biology, Ufa Federal Research Centre, RAS, 450054 Ufa, 10 Russia; ilshat.rafkatovitch@gmail.com (I.G.); g.v.sharipova@mail.ru (G.S.); veselov@anrb.ru (D.V.) 11
3 Laboratory of agrobiology, Ufa Institute of Biology, Ufa Federal Research Centre, RAS, 450054 Ufa, Russia; 12 che-kov@mail.ru (S.C.)"
--> The author should discuss with the journal whether the journal will accept email addresses for all members who are both authors and co-authors of the article.
4. Abstract: The abstract should be written more precisely and explain novelty of this work; Please check throughout the manuscript that abbreviations/acronyms are defined the first time they appear in each of three sections: the abstract; the main text; the first figure or table.
The summary should be written in between 5 and 8 sentences to ensure that the core content of the experiment is conveyed to the reader.
For example:
Sentences 1 and 2: Introduce the research and the reason for choosing the topic (Please note that the novelty of the research is very important, problem statement).
Sentence 3: What research tools and methods do you use in this article? (indication of methodology)
Sentences 4 to 7: Main results of the research, write 1 sentence for each result (main findings)
Sentence 8: Conclusion and recommendations (principle conclusion).
5. The discussion results as well as the conclusion should be rewritten to closely follow the research content and the experiment's objectives. The discussion results should be divided into subsections and focused on the main research contents of the experiment (each subsection is a main content of the experiment).
The conclusion closely follows the results obtained from the main content above

Author Response
Response to Reviewer 1 Comments
Point 1:
Table 2. Comparative evaluation of the morphometric parameters of Secale cereale L. plants under conditions of soil pollution with petroleum and with the use of the "Lenoil" preparation à italicize the name of the species
Table 2. Changes in the morphological structure of the roots of 30-day-old rye plants under conditions of soil contamination with petroleum and treatment with "Lenoil" preparation need to renumber these 2 tables
Response 1: The titles and numbers of Tables 1 and 2 were changed.
Point 2: Title: “Influence of a Hydrocarbons Biodestructor on the Growth and 2 Content of Phytohormones in Secale Cereale L. Plants under Oil 3 Pollution of the Soil”
--> The capitalization of the first character of a letter is a rule of the journal or not, if it is a rule of the journal, it is okay, but if it is not the rule of the journal, the author should limit the capitalization of first characters in the title of the article.
Response 2: The analysis of the article was carried out and adjustments were made
Influence of a Hydrocarbons Biodestructor on the Growth and Content of Phytohormones in Secale cereale L. Plants under Petroleum Pollution of the Soil
Point 3. The author should discuss with the journal whether the journal will accept email addresses for all members who are both authors and co-authors of the article.
Response 3: Issued according to the requirements of the journal
Point 4.
Abstract: The abstract should be written more precisely and explain novelty of this work; Please check throughout the manuscript that abbreviations/acronyms are defined the first time they appear in each of three sections: the abstract; the main text; the first figure or table.
The summary should be written in between 5 and 8 sentences to ensure that the core content of the experiment is conveyed to the reader.
For example:
Sentences 1 and 2: Introduce the research and the reason for choosing the topic (Please note that the novelty of the research is very important, problem statement).
Sentence 3: What research tools and methods do you use in this article? (indication of methodology)
Sentences 4 to 7: Main results of the research, write 1 sentence for each result (main findings)
Sentence 8: Conclusion and recommendations (principle conclusion).
Response 4
Sentences 1 and 2: Introduce the research and the reason for choosing the topic (Please note that the novelty of the research is very important, problem statement).
Response: These sentences have been changed: “The phytoremediation of soil contaminated with petroleum depends on association of plants with rhizosphere bacteria capable of promoting plant growth and destroying petroleum hydrocarbonates. In the present work we studied effects of “Lenoil” biological product containing bacteria Pseudomonas turukhanskensis IB 1.1 capable of destroying petroleum hydrocarbons on rye plants which previously proved to be resistant to weak oil pollution of gray forest soil and on composition of microorganisms in their rhizosphere.”
Sentence 3: What research tools and methods do you use in this article? (indication of methodology)
Response: The sentence was changed: “Composition of microorganisms in the rhizosphere of rye roots was studied, morphometric parameters of shoots and roots of rye plants were estimated and hormone concentration was immunoassayed under conditions of 4% petroleum pollution of the soil.”
Sentences 4 to 7: Main results of the research, write 1 sentence for each result (main findings)
Response: The sentence was changed: “Addition of petroleum to the soil, increased the content of oligonitrophilic (by 24%) and hydrocarbon-oxidizing (by 33%) microorganisms, but the content of cellulolytic (by 12.5 times) microorganisms in the rhizosphere decreased. The use of Lenoil led to further increase in the number of cellulolytic (by 5.6 times) and hydrocarbon-oxidizing (by 3.8 times) microorganisms and a decrease in the number of oligonitrophilic (by 22.7%) microorganisms in the rhizosphere. Under petroleum pollution, the content of IAA, zeatin riboside, zeatin nucleotide, and zeatin decreased, while the content of ABA increased in the shoots roots of rye plants. Leniol-treatment led to an 8-fold increase in the IAA content in the roots and a decrease in the ABA content in the aerial part and in the roots. It was shown for the first time that the treatment of petroleum-contaminated soil with the "Lenoil" increased root mass due to the development of lateral roots, concurrent with high root IAA content. Petroleum pollution increased deposition of lignin and suberin in the roots, which strengthened the apoplastic barrier and thus reduced the infiltration of toxic components. The deposition of suberin and lignin decreased under 'Lenoil' treatment indicating a decrease in the concentration of toxic petroleum components in the soil degraded by the bacteria.”
Sentence 8: Conclusion and recommendations (principle conclusion).
Response: Conclusion was changed: “Thus, the biological preparation reduced the growth-inhibiting effect of petroleum on rye plants by increaseing the content of growth-stimulating phytohormones (IAA and cytokinins), and reducing the content of ABA justifying importance of further studying of hormones for improvement of phytoremediation.
Point 5. The discussion results as well as the conclusion should be rewritten to closely follow the research content and the experiment's objectives. The discussion results should be divided into subsections and focused on the main research contents of the experiment (each subsection is a main content of the experiment).
The conclusion closely follows the results obtained from the main content above
Response 5: The text of the conclusion was specified and contains the main results of the study. Journal rules allow the absence of division of results and discussions into subsections
According to the guidelines of the journal, results and their discussion were merged into one section. We followed recommendation of respected reviewer by dividing it into subsections: 3.1. Composition of microorganisms in the rhizosphere of rye roots; 3.2. Morphometric parameters of shoots and roots of rye plants under soil pollution and “Lenoil” treatment; 3.3. Concentration of hormones in rye plants under soil pollution and treatment with “Lenoil” and 3.4. Root structure of rye plants under soil pollution and treatment with “Lenoil”.
- Conclusions
The results of the present research are important for understanding plant-microbial interactions and using the results in the restoration of petroleum-contaminated soils with phytoremediants and bacterial preparations. The application of the microbiological preparation “Lenoil” containing hydrocarbon-oxidizing microorganisms of the Pseudomonas genus led to an amelioration of living conditions for rye plants resulting in improved morphometric parameters of rye plants. The effects of “Lenoil” can be due not only to the presence of Pseudomonas microorganisms in the product itself, but also to its effect on composition of microorganisms in the soil. The concentration of auxins and cytokinins increased in plants treated with Lenoil, in accordance with the previously shown ability of microorganisms of this biological product to synthesize these hormones [2]. There was a decrease in the content of the stress phytohormone, ABA, both in the aerial part and in the roots of Lenoil-treated plants. The content of IAA increased in both parts of the plant, but to a greater extent in the roots which stimulated their growth. The rapid increase in the length of the aerial part was associated with an increase of zeatin content. In plants of this variant, the length of the aerial parts and roots were close to the values of the control variant grown in the absence of petroleum contamination. Thus, under the conditions of petroleum pollution and soil treatment with the “Lenoil”, we showed the effect of the biological product on the state of the endogenous hormonal system of plants, which led to the activation of growth processes and a change in the structure of the root cell wall. Thus estimation of the content of IAA, cytokinins, and ABA is important for selecting effective concentrations of biological preparations and creating their combinations during phytoremediation activities.

Reviewer 2 Report
Reviewed paper presents effects of 'Lenoil' preparation on microorganisms and Secale cereale growth under petroleum pollution. Obtained results gave basic conclusion which could be continued in more detailed works.
Main comments:
- In 'Materials and Methods' information about plants is missing
- In first Table 2 (which should be Table 1) I suppose that data for shoots are wrong because they are duplicated (same values for length and fresh weight)
- Statistics are missing (Fig.1 and 2) or presumably poorly performed (Table 1)
- The conclusions are of poor quality, please improve
Detailed comments are included in PDF file

Author Response
Response to Reviewer 2 Comments
Point 1: In 'Materials and Methods' information about plants is missing
Response 1: The information about rye has been added.
Five plants of Secale cereale L. (cv Tatiana) were grown in 0.5-liter pots at a 12-hour light period, PAR was 535 µmol s-1 m-2, and an air temperature of 22–25°C. Soil moisture was maintained throughout the experiment at 60% of full capacity.
Point 2:
In first Table 2 (which should be Table 1) I suppose that data for shoots are wrong because they are duplicated (same values for length and fresh weight)
Response 2: Аdjustments have been made
Unfortunately, incorrect data on the shoot weight were allowed due to incorrect copying of the text.
Table 1. Comparative evaluation of the morphometric parameters of Secale cereale L. plants under conditions of soil pollution with petroleum and with the use of the "Lenoil" preparation
Plant part |
Parameter |
Control |
Petroleum |
Petroleum + “Lenoil” |
Shoot |
Length, cm |
17,83a±1,22 |
8,46b±0,52 |
32,11c±1,32 |
Fresh weight, g |
8,36a±0,66 |
2,38b±0,26 |
10,46a±0,96 |
|
Root |
Length, cm |
12,24a±1,04 |
7,34b±0,56 |
18,41c±1,26 |
Fresh weight, g |
1,75a±0,12 |
0,47b±0,04 |
3,82c±0,22 |
Data are means ± SE. Significantly different means are marked with different letters at p ≤ 0.05 (ANOVA followed by Duncan’s test).
Point 3. Statistics are missing (Fig.1 and 2) or presumably poorly performed (Table 1)
Response 3: Statistical analysis data are entered in figures 1 and 2 and table 1
Point 4. The conclusions are of poor quality, please improve
Response 4: Conclusion modified in accordance with the recommendations of the reviewer
- Conclusions
The results of the present research are important for understanding plant-microbial interactions and using the results in the restoration of petroleum-contaminated soils with phytoremediants and bacterial preparations. The application of the microbiological preparation “Lenoil” containing hydrocarbon-oxidizing microorganisms of the Pseudomonas genus led to an amelioration of living conditions for rye plants resulting in improved morphometric parameters of rye plants. The effects of “Lenoil” can be due not only to the presence of Pseudomonas microorganisms in the product itself, but also to its effect on composition of microorganisms in the soil. The concentration of auxins and cytokinins increased in plants treated with Lenoil, in accordance with the previously shown ability of microorganisms of this biological product to synthesize these hormones [2]. There was a decrease in the content of the stress phytohormone, ABA, both in the aerial part and in the roots of Lenoil-treated plants. The content of IAA increased in both parts of the plant, but to a greater extent in the roots which stimulated their growth. The rapid increase in the length of the aerial part was associated with an increase of zeatin content. In plants of this variant, the length of the aerial parts and roots were close to the values of the control variant grown in the absence of petroleum contamination. Thus, under the conditions of petroleum pollution and soil treatment with the “Lenoil”, we showed the effect of the biological product on the state of the endogenous hormonal system of plants, which led to the activation of growth processes and a change in the structure of the root cell wall. Thus estimation of the content of IAA, cytokinins, and ABA is important for selecting effective concentrations of biological preparations and creating their combinations during phytoremediation activities.

Reviewer 3 Report
The present work was interesting; however, the authors should address the following concerns raised by the reviewer before this paper can be considered for publication.
1. Abstract should be rearranged. The first sentence should describe the background of the experiment and the last sentence should strengthen the innovation of the present work. When describing the results, quantifiable data should be added. For instance, "treatment A significantly increased the IAA content by X% compared with control".
2. Introduction part was too short. The authors should describe the role of phytohormones as well as microorganism in petroleum pollution in more detail.
3. Line 60: The soil layer information was missing. The chemical condition of the soil was not accurate enough, e.g. the soil pH was between 5.0 to 6.0. Did the authors measure the soil properties?
4. Line 69: The cultivar of the plant species was missing.
5. Line 109, 152, 182, 200, and 227: "P" should be italic.
6. Line 180: Table 2 should be revised to "Table 1".
7. In the figure legends of Figure 1 and 2, the authors mentioned that significant different means are marked with different letters; however, no letters were observed in these figures.
8. In results part of the manuscript, the data should be presented in quantifiable style as mentioned above in Abstract.
I recommend revising the manuscript by native English speakers. There are a lot of grammar mistakes throughout the whole manuscript.
Author Response
Response to Reviewer 3 Comments
Point 1: Abstract should be rearranged. The first sentence should describe the background of the experiment and the last sentence should strengthen the innovation of the present work. When describing the results, quantifiable data should be added. For instance, "treatment A significantly increased the IAA content by X% compared with control".
Response 1: The structure and content of the abstract have been changed in accordance with the recommendations of the reviewer.
Abstract: The phytoremediation of soil contaminated with petroleum depends on association of plants with rhizosphere bacteria capable of promoting plant growth and destroying petroleum hydrocarbonates. In the present work we studied effects of “Lenoil” biological product containing bacteria Pseudomonas turukhanskensis IB 1.1 capable of destroying petroleum hydrocarbons on Secale cereale L plants which previously proved to be resistant to weak oil pollution of gray forest soil and on composition of microorganisms in their rhizosphere. Composition of microorganisms in the rhizosphere of rye roots was studied, morphometric parameters of shoots and roots of rye plants were estimated and hormone concentration was immunoassayed under conditions of 4% petroleum pollution of the soil. Addition of petroleum to the soil, increased the content of oligonitrophilic (by 24%) and hydrocarbon-oxidizing (by 33%) microorganisms, but the content of cellulolytic (by 12.5 times) microorganisms in the rhizosphere decreased. The use of Lenoil led to further increase in the number of cellulolytic (by 5.6 times) and hydrocarbon-oxidizing (by 3.8 times) microorganisms and a decrease in the number of oligonitrophilic (by 22.7%) microorganisms in the rhizosphere. Under petroleum pollution, the content of IAA, zeatin riboside, zeatin nucleotide, and zeatin decreased, while the content of ABA increased in the shoots roots of rye plants. Leniol-treatment led to an 8-fold increase in the IAA content in the roots and a decrease in the ABA content in the aerial part and in the roots. It was shown for the first time that the treatment of petroleum-contaminated soil with the "Lenoil" increased root mass due to the development of lateral roots, concurrent with high root IAA content. Petroleum pollution increased deposition of lignin and suberin in the roots, which strengthened the apoplastic barrier and thus reduced the infiltration of toxic components. The deposition of suberin and lignin decreased under 'Lenoil' treatment indicating a decrease in the concentration of toxic petroleum components in the soil degraded by the bacteria. Thus, the biological preparation reduced the growth-inhibiting effect of petroleum on rye plants by increaseing the content of growth-stimulating phytohormones (IAA and cytokinins), and reducing the content of ABA justifying importance of further studying of hormones for improvement of phytoremediation.
Point 2: 2. Introduction part was too short. The authors should describe the role of phytohormones as well as microorganism in petroleum pollution in more detail.
Response 2: The role of phytohormones and microorganisms is considered in more detail in the course of the discussion of the experimental material. In this regard, in order not to duplicate the data, they are not contained in the text of the introduction.
Point 3. Line 60: The soil layer information was missing. The chemical condition of the soil was not accurate enough, e.g. the soil pH was between 5.0 to 6.0. Did the authors measure the soil properties?
Response 3: The average soil pH is 5.6±0.4.
The experiments were carried out under laboratory conditions using grey forest soil collected in the north-eastern part of the Ufa district of the Republic of Bashkortostan. The soil was ground up and sifted through 3 mm mesh. The soil had the following agrochemical indicators: clay-illuvial agro-chernozem, characterized by medium humus content (4.4±0,5%) and slight acid reaction (pH 5.6±0,4).
Point 4. Line 69: The cultivar of the plant species was missing.
Response 4: The name of the species and cultivar of the plant was included in the text of the article
Five plants of Secale cereale L. (cv Tatiana) were grown in 0.5-liter pots at a 12-hour light period, PAR was 535 µmol s-1 m-2, and an air temperature of 22–25°C. Soil moisture was maintained throughout the experiment at 60% of full capacity. Commercial petroleum was added to the soil at a concentration of 4% of the dry weight of the soil. After 30 days, 0.3 ml of "Lenoil" per 100 g of dry soil was added to half of the vessels containing petroleum. Plants were sown 30 days after the addition of the biopreparation. Plants grown in soil not contaminated with petroleum served as control.
Point 5. Line 109, 152, 182, 200, and 227: "P" should be italic.
Response 5: It was fixed
Point 6. Line 180: Table 2 should be revised to "Table 1".
Response 6: It was fixed
Table 1. Comparative evaluation of the morphometric parameters of Secale cereale L. plants under conditions of soil pollution with petroleum and with the use of the "Lenoil" preparation
Table 2. Changes in the morphological structure of the roots of 30-day-old Secale cereale L. plants under conditions of soil contamination with petroleum and treatment with "Lenoil" preparation
Point 7. In the figure legends of Figure 1 and 2, the authors mentioned that significant different means are marked with different letters; however, no letters were observed in these figures.
Response 7: Statistical analysis data are included in Figs. 1 and 2
Point 8. In results part of the manuscript, the data should be presented in quantifiable style as mentioned above in Abstract.
Response 8: The text of the conclusion and annotation was corrected in accordance with the recommendations of the reviewer

Reviewer 4 Report
This paper examined the influence of the biological preparation "Lenoil" on the growth and balance of phytohormones of Secale cereale plants under conditions of soil petroleum pollution. The growth, hormones and anatomy of plants were determined, as well as soil microorganisms. The results showed that application of "Lenoil" changed the quantitative and qualitative composition of cellulolytic, oligonitrophilic, and hydrocarbon-oxidizing microorganisms in the rhizosphere. Biological product reduced the growth-inhibiting effect of petroleum on plants and increase growth-stimulating hormones. Deposition of suberin and lignin was decreased by "Lenoil" treatment. Many results got, but the idea was not that novel and the explanation and the discussion needs further improved. Some specific comments were below:
1. Abstract:more data could be included.
2. Introduction:more similar researches needs to be added. The novelty of your research should be emphasized.
3. Material and methods: The content of the petroleum should be supplied. How many plants per pot? The name of the plant? How many lenoil added per pot?
4. Results and discussion:the data of the heterotrophic microorganisms, the number of micromycetes should be provided.
5. The significant different was not shown in figure 1. The data of the fresh weight of the shoot was wrong in table 2. Line 239, ZN not GN.
6. The details of the Pseudomonas should be added such as the ability of synthesizing significant concentrations of IAA and cytokinins, the ability of destructing hydrocarbons of petroleum.
7. The reason of testing the ZN,ZR and Z? What’s the relationship between these indexes with other parameters?
8. The changes of the petroleum content in soils and in plants could be added to further support the discussion of the results.
Author Response
Response to Reviewer 4 Comments
Point 1: 1. Abstract:more data could be included.
Response 1: The structure and content of the abstract has been changed in accordance with the recommendations of the reviewer.
Abstract: The phytoremediation of soil contaminated with petroleum depends on association of plants with rhizosphere bacteria capable of promoting plant growth and destroying petroleum hydrocarbonates. In the present work we studied effects of “Lenoil” biological product containing bacteria Pseudomonas turukhanskensis IB 1.1 capable of destroying petroleum hydrocarbons on Secale cereale L plants which previously proved to be resistant to weak oil pollution of gray forest soil and on composition of microorganisms in their rhizosphere. Composition of microorganisms in the rhizosphere of rye roots was studied, morphometric parameters of shoots and roots of rye plants were estimated and hormone concentration was immunoassayed under conditions of 4% petroleum pollution of the soil. Addition of petroleum to the soil, increased the content of oligonitrophilic (by 24%) and hydrocarbon-oxidizing (by 33%) microorganisms, but the content of cellulolytic (by 12.5 times) microorganisms in the rhizosphere decreased. The use of Lenoil led to further increase in the number of cellulolytic (by 5.6 times) and hydrocarbon-oxidizing (by 3.8 times) microorganisms and a decrease in the number of oligonitrophilic (by 22.7%) microorganisms in the rhizosphere. Under petroleum pollution, the content of IAA, zeatin riboside, zeatin nucleotide, and zeatin decreased, while the content of ABA increased in the shoots roots of rye plants. Leniol-treatment led to an 8-fold increase in the IAA content in the roots and a decrease in the ABA content in the aerial part and in the roots. It was shown for the first time that the treatment of petroleum-contaminated soil with the "Lenoil" increased root mass due to the development of lateral roots, concurrent with high root IAA content. Petroleum pollution increased deposition of lignin and suberin in the roots, which strengthened the apoplastic barrier and thus reduced the infiltration of toxic components. The deposition of suberin and lignin decreased under 'Lenoil' treatment indicating a decrease in the concentration of toxic petroleum components in the soil degraded by the bacteria. Thus, the biological preparation reduced the growth-inhibiting effect of petroleum on rye plants by increaseing the content of growth-stimulating phytohormones (IAA and cytokinins), and reducing the content of ABA justifying importance of further studying of hormones for improvement of phytoremediation.
Point 2: Introduction:more similar researches needs to be added. The novelty of your research should be emphasized.
Response 2: The role of phytohormones and microorganisms is considered in more detail in the course of the discussion of the experimental material. In this regard, in order not to duplicate the data, they are not contained in the text of the introduction.
Point 3. Material and methods: The content of the petroleum should be supplied. How many plants per pot? The name of the plant? How many lenoil added per pot?
Response 3: We have added to the text:
The petroleum content in the soil was 4%.
5 rye plants grew in a pot
Plant name - Secale cereale L. added to text
“Lenoil” (titer not less than 1×108 CFU g-1) was applied at the rate of 0.3 ml per 100 g of dry soil
Five plants of Secale cereale L. (cv Tatiana) were grown in 0.5-liter pots at a 12-hour light period, PAR was 535 µmol s-1 m-2, and an air temperature of 22–25°C. Soil moisture was maintained throughout the experiment at 60% of full capacity. Commercial petroleum was added to the soil at a concentration of 4% of the dry weight of the soil. After 30 days, 0.3 ml of "Lenoil" per 100 g of dry soil was added to half of the vessels containing petroleum. Plants were sown 30 days after the addition of the biopreparation. Plants grown in soil not contaminated with petroleum served as control.
Point 4
Results and discussion:the data of the heterotrophic microorganisms, the number of micromycetes should be provided.
Response 4:
Our assessment of the indicator of the state of soil microbiocenosis showed that in the control variant, the total number of heterotrophic microorganisms was about 9·106 CFU g-1 of soil in the rhizosphere of rye plants. Under the influence of the pollutant, this indicator decreased by 63%, which could be due to the direct toxic effect of petroleum hydrocarbons.
The use of the "Lenoil" preparation increased the number of heterotrophic microorganisms up to about 3·107 CFU g-1 of soil (compared to 3·106 CFU g-1 of contaminated soil). This group of microorganisms characterizes the total microbial number and the change in the number depends on the proportion of sensitive representatives of the microbiome both in whole soil and in the plant rhizosphere [20,21]. It is known that the treatment with the biopreparation improves living conditions of oil-contaminated soils for both plants and rhizospheric microorganisms [2].
There was also a significant growth of microscopic fungi, both in petroleum-contaminated and "Lenoil"-treated soil. In the presence of petroleum in the rye rhizosphere, the number of micromycetes increased about 20 times compared with the control (from 1·103 to 22·103 propagules g-1 of soil), and after the addition of the biological preparation – by another 2 times (up to 46.75·103 propagule g-1 soil).
Point 5. The significant different was not shown in figure 1. The data of the fresh weight of the shoot was wrong in table 2. Line 239, ZN not GN.
Response 5: It's been fixed
Table 1. Comparative evaluation of the morphometric parameters of Secale cereale L. plants under conditions of soil pollution with petroleum and with the use of the "Lenoil" preparation
Plant part |
Parameter |
Control |
Petroleum |
Petroleum + “Lenoil” |
Shoot |
Length, cm |
17,83a±1,22 |
8,46b±0,52 |
32,11c±1,32 |
Fresh weight, g |
8,36a±0,66 |
2,38b±0,26 |
10,46a±0,96 |
|
Root |
Length, cm |
12,24a±1,04 |
7,34b±0,56 |
18,41c±1,26 |
Fresh weight, g |
1,75a±0,12 |
0,47b±0,04 |
3,82c±0,22 |
Point 6
The details of the Pseudomonas should be added such as the ability of synthesizing significant concentrations of IAA and cytokinins, the ability of destructing hydrocarbons of petroleum.
Response 6: In our work, we used the data obtained by our colleagues on the ability to degrade oil and the ability to synthesize phytohormones. There is a reference in the text to the relevant work. We were interested in plant-microbial interaction under conditions of rye plant growth under soil contamination.
Point 7. The reason of testing the ZN,ZR and Z? What’s the relationship between these indexes with other parameters?
Response 7: The analysis of zeatin derivatives makes it possible to judge the nature of changes in the hormonal response. For example, the accumulation of ZN indicates a decrease in the supply of zeatin to the aerial part of the plant, the accumulation of ZR indicates an increase in the supply of the transport form from the roots to the aerial part, and so on.
Point 8. The changes of the petroleum content in soils and in plants could be added to further support the discussion of the results.
Response 8: Data on the decrease in oil in the soil are contained in lines 113 and 114, the determination of the oil content inside the plant was not carried out.
The determination of the petroleum content in the soil showed that in the variant with application of the "Lenoil" microbiological preparation (30 days without plants plus 30 days with rye plants), we obtained a decrease in its content in the soil from 4 to 2.4% after first 30 days and after another 30 days in the presence of plants - down to 1.6%. In the variant without soil treatment with the preparation, petroleum content was found to be 2.8% after 2 months of the experiment.

Round 2
Reviewer 2 Report
Thank you for considering all my comments.
I found 1 typo in line 38 - 'increaseing'
Please add short information about statistics to each Figure because it is hard to follow it, eg. In Figure 1 I do not understand why for CLB there are not any differences (only a) and for HOM 'Control' does not differ from 'Petroleum+Lenoil'
In my opinion the manuscript is well-written
Author Response
Point 1: The typo in line 38 - 'increaseing'
Response 1: We are grateful to the respected reviewer for carefully reading our article and finding the typo
Point 2. Please add short information about statistics to each Figure because it is hard to follow it, eg. In Figure 1 I do not understand why for CLB there are not any differences (only a) and for HOM 'Control' does not differ from 'Petroleum+Lenoil'
Response 2: We apologize for the incorrect statistics in Fig. 1, the wrong designation was indicated there. We've made adjustments. We have also checked other figures and tables

Reviewer 3 Report
Dear authors, thank you for the revisions.
Author Response
Point 1: Dear authors, thank you for the revisions.
Response 2: We are deeply grateful to the reviewer for his work and for his recommendations to improve the quality of the English language.

Reviewer 4 Report
The authors have made corresponding revisions. The novelty could be further illustrated in the introduction. And some information in the response could also be further supplied in the paper, e.g. point 6, point 7. the Response 8: Data on the decrease in oil in the soil are contained in lines 123 and 124 not 113 and 114.
Author Response
Point 1: The authors have made corresponding revisions. The novelty could be further illustrated in the introduction. And some information in the response could also be further supplied in the paper, e.g. point 6, point 7.
“Response 6: In our work, we used the data obtained by our colleagues on the ability to degrade oil and the ability to synthesize phytohormones. There is a reference in the text to the relevant work. We were interested in plant-microbial interaction under conditions of rye plant growth under soil contamination.
Response 7: The analysis of zeatin derivatives makes it possible to judge the nature of changes in the hormonal response. For example, the accumulation of ZN indicates a decrease in the supply of zeatin to the aerial part of the plant, the accumulation of ZR indicates an increase in the supply of the transport form from the roots to the aerial part, and so on.”
Response 1: In accordance with your wishes, we made a clarification about the novelty of our work in the introduction and outlined the tasks of cytokinin forms studying in the discussion of the results.
Point 2: Response 8: Data on the decrease in oil in the soil are contained in lines 123 and 124 not 113 and 114.
Response 2: You are right, unfortunately, we didn't check the line numbers after putting all the text corrections
